# PI Film Laser Micro-Cutting for Quantitative Manufacturing of Contact Spacer in Flexible Tactile Sensor

**DOI:** 10.3390/mi12080908

**Published:** 2021-07-30

**Authors:** Congyi Wu, Tian Zhang, Yu Huang, Youmin Rong

**Affiliations:** 1State Key Lab of Digital Manufacturing Equipment and Technology, Huazhong University of Science and Technology, Wuhan 430070, China; cyw@hust.edu.cn (C.W.); tianz@hust.edu.cn (T.Z.); 2School of Mechanical Science and Engineering, Huazhong University of Science and Technology, Wuhan 430070, China

**Keywords:** laser micro-cutting, PI film, contact spacer, tactile sensor

## Abstract

The contact spacer is the core component of flexible tactile sensors, and the performance of this sensor can be adjusted by adjusting contact spacer micro-hole size. At present, the contact spacer was mainly prepared by non-quantifiable processing technology (electrospinning, etc.), which directly leads to unstable performance of tactile sensors. In this paper, ultrathin polyimide (PI) contact spacer was fabricated using nanosecond ultraviolet (UV) laser. The quality evaluation system of laser micro-cutting was established based on roundness, diameter and heat affected zone (HAZ) of the micro-hole. Taking a three factors, five levels orthogonal experiment, the optimum laser cutting process was obtained (pulse repetition frequency 190 kHz, cutting speed 40 mm/s, and RNC 3). With the optimal process parameters, the minimum diameter was 24.3 ± 2.3 μm, and the minimum HAZ was 1.8 ± 1.1 μm. By analyzing the interaction process between nanosecond UV laser and PI film, the heating-carbonization mechanism was determined, and the influence of process parameters on the quality of micro-hole was discussed in detail in combination with this mechanism. It provides a new approach for the quantitative industrial fabrication of contact spacers in tactile sensors.

## 1. Introduction

Flexible tactile sensors are widely used in electronic skin [1,2,3], robots [4,5,6], wearable devices [7,8,9], etc. Depending on the source of the signal, the tactile sensors are classified into capacitive-type [10,11], piezoelectric-type [12,13], triboelectric-type [14] and piezoresistive-type [15,16,17,18]. Among them, the piezoresistive tactile sensor has been extensively studied for its advantages of simple structure, low manufacturing cost and convenient signal processing. As one of the core components in piezoresistive tactile sensors, a contact spacer acts as a regulator of sensitivity and test range. For instance, a 2 μm thick rigid SiO_2_ layer was applied as contact spacer and the sensor achieved ultra-high sensitivity of about 100–970 μA/kPa, but inflexibility limits its use scope [19]. In addition, a polyvinyl pyrrolidone (PVP) nanowire mesh was used to isolate the silver-plated micro-pyramids, creating a tactile sensor with an ultra-high sensitivity and ultra-wide detection range [20]. A tactile sensor with adjustable sensitivity and test range was fabricated by isolating the wrinkled polypyrrole (PVP) film through a polyvinyl alcohol (PVA) nanowire mesh [21]. However, the insulating nanowire mesh prepared by electrospinning process not only cannot quantify the size of single micro-hole, but also cannot uniformize the size of multiple micro-holes. The quantitative manufacturing of contact spacers thus should meet the following conditions: (1) flexible insulating film to isolate resistance, (2) ultra-thin film (1–10 μm) to form micro-sized peak in the micro-hole during compression deformation and (3) 5–100 μm diameter micro-hole can be fabricated quantitatively to form large contact resistance in a single micro-hole during compression deformation.

Polyimide (PI) film has many excellent properties such as flexibility, ultra-thin manufacturing (≥3 μm), insulation, high temperature resistance, radiation-resistance, etc., making it an excellent raw material for contact spacers. Ultra-thin PI film thus can be used to fabricate the contact spacer, and traditional machining methods such as stamping cannot fabricate micro-holes in the film. Laser-based micro-nano manufacturing has been widely used in various industries, such as metal organic framework (MOF) additive manufacturing [22], polydimethylsiloxane (PDMS) laser cutting in flexible electronics, laser direct writing for interdigitated electrode [23,24], laser induced biodegradation [25,26,27], etc. [28], laser micro-cutting thus may be a better method for preparing micro-holes through PI films. Nanosecond lasers are widely used in industrial production due to their low cost and high stability [29,30]. Ultraviolet (UV, 355 nm) laser has lower thermal ablation effect on the substrate than other wavelengths of light, so it is more suitable for manufacturing of organic materials to obtain smaller heat affected zones (HAZs) [31]. However, whether the interaction mechanism between the PI film and the UV nanosecond laser is a photothermal process or photochemical process, or both, remains to be further investigated [32,33,34]. In addition, the mechanism of PI film decomposition under laser irradiation conditions is still unclear, and the relationship between the thermal damage and the process parameters needs to be further investigated.

In this paper, nanosecond UV laser was chosen to fabricate the ultra-thin micro-hole PI film contact spacer. However, the evaluation index of micro-holes quality, the influence of process parameters on the evaluation index, and the interaction mechanism between UV laser and PI films are yet to be investigated. Firstly, an evaluation system of micro-hole based on roundness, diameter and size of the heat affected zone (HAZ) was established by analyzing the morphology after the laser micro-cutting. Secondly, the influence of process parameters was investigated by a 3-factor, 5-level orthogonal experiment to obtain the optimal process parameters. Finally, the interaction mechanism based on heating-carbonization was confirmed through the analysis of the products after laser micro-cutting and used to analyze the influence of process parameters. It provides a new approach for the quantitative industrial fabrication of contact spacers in tactile sensors.

## 2. Experiment Description

### 2.1. Laser Cutting System

As shown in Figure 1, the laser cutting system consisted of an industrial personal computer (IPC), nanosecond UV laser, extender lens, scan mirror, focusing lens and support platform. The nanosecond UV laser (Poplar-355-15A5, Huaray Precision Laser, Wuhan, China) had a maximum power of 12 W and pulse width < 15 ns. The extender lens was used to reduce laser processing power, because the diameter of laser spot expanded by the expander lens was larger than the entrance diameter of the scan mirror. The scan mirror (S10-355-D, SCANLAB, Pulheim, Germany) was used to control laser cutting path. The focusing lens (LINOS4401-402-000-20, QIOPTIQ, Gottingen, Germany) used in this study had a focal length of 167 mm. The 5 μm ultra-thin PI film (Taobao online store) was pasted on a piece of stainless steel, which was fixed on a motion platform. Output power of laser can be adjusted by setting pulse repetition frequency (PRF) values, and the correspondent relationship is shown in Table 1, where the power values are measured by power meter. After the laser processing, each sample was ultrasonically cleaned in deionized water and ethanol solution for 10 min in sequence, and then dried at room temperature.

### 2.2. Experiment Design of Laser Cutting Micro-Holes

According to previous experiments, the quality of micro-hole is mainly determined by the following three process parameters: PRF, cutting speed (CS) and repetition number of cuts (RNC). The nanosecond UV laser’s PRF could be varied from 50 to 200 kHz, the CS could be varied from 0.1 to 10,000 mm/s, and the RNC could be increased indefinitely. Higher cutting powers, slower cutting speeds and more RNC help to remove material but can also lead to more defects such as large HAZ. Therefore, based on preliminary test results, the specific PRF value varied from 160 to 200 kHz, the CS value varied from 10 to 50 mm/s, and the RNC value varied from 1 to 5. Furthermore, each parameter was set to 5 arithmetic values within the range of variation, and their influences on the micro-cutting quality was evaluated through orthogonal experiment of 3 factors and 5 levels. The values and levels of the main process parameters are shown in Table 2, and the specific process parameters are shown in Table 3.

### 2.3. Characterization

The basic morphology of the micro-hole was observed by a laser scanning confocal microscope (TCS SP8 X, Leica, Weztlar, Germany). In addition, a field emission scanning electron microscope (FS-EM, FEI Sirion 200, Santa Clara, CA, USA) was used to observe more detailed morphology and component analysis. The Raman spectra were carried out on a Raman spectrometer (HORIBA Jobin Yvon, Paris, France) with a 30 mW He-Cd laser of 532 nm. The thermogravimetric analysis (TGA) and differential thermal analysis (DTA) were measured by synchronous thermal analyzer (STA PT1600, LINSEIS, Zerb, Germany), that was carried out under air flow from room temperature to 800 °C at 10 min^−1^.

## 3. Results and Discussion

### 3.1. Quality Evaluation System of Laser Micro-Cutting

Firstly, the evaluation system of micro-hole quality was established based on morphological characteristics, and the eigenvalues of each sample in the orthogonal experiment were extracted based on this system. Secondly, the influence of process parameters on micro-hole quality was discussed by orthogonal analysis of the eigenvalues of each sample. As shown in Figure 2, the typical micro-hole is not a perfect circle, and inscribed circle of micro-hole d_1_ and circumscribed circle of micro-hole d_2_ can be drawn. In addition, circumscribed circle d_3_ can be drawn outside the micro-hole due to the presence of the HAZ. From the three-dimensional image of micro-hole measured by a laser scanning confocal microscope, it can be judged whether the micro-hole is through-hole (TH). According to the values of d_1_, d_2_, and d_3_, evaluation indexes of the roundness, the diameter of the micro-hole, and the size of the HAZ can be determined, and the calculation formulas are as follows:(1)$$\mathrm{Roundness}=\frac{d_{2}-d_{1}}{2}$$
(2)$$\mathrm{Diameter}=\frac{d_{1}+d_{2}}{2}$$
(3)$$\mathrm{Width}\;\mathrm{of}\;\mathrm{HAZ}=\frac{d_{3}-\mathrm{Dia}}{2}$$

Based on this evaluation system, eight samples were selected in each parameter for the calculation of the evaluation indexes. Sample values of d_1_, d_2_, d_3_ were obtained by averaging five measurements after removing maximum and minimum values. In addition, it was determined that the cutting parameter can cut the PI film only when all the micro-holes were through-holes. The typical micro-hole morphologies of each parameter are shown in Figure 3, and the statistical data are shown in Table 4.

### 3.2. Variance Analysis of the L_25_(5^3^) Orthogonal Experiments

Range is the difference between the maximum values and the minimum values of the experimental results. Range analysis can quickly determine the optimal level of a single factor and the optimal level combination of multiple factors in an orthogonal experiment. When the evaluation index is the roundness, the results of the range analysis of each parameter parameter are shown in Table 5 and Figure 4a. The parameter parameter PRF has a range R_1_ of 0.706 μm for the evaluation index R. As the value of the PRF increases, the roundness value decreases continuously, and the best R is obtained at lever 5 (200 kHz). The range R_2_ and R_3_ of the process parameters CS and RNC are 0.681 and 0.201 μm, respectively. As the values of CS and RNC increase, the value of R also decreases, and the roundness is optimal at a cutting speed level of 5 (50 mm/s) and a cutting number level of 5 (5). Therefore, in this orthogonal experiment, the optimal R can be obtained when the parameter parameter combination is 5-5-5. As shown in Table 4, the through-hole can be obtained when the parameter combination is 5 (200 kHz) - 5 (50 mm/s) - 4 (4). Therefore, for the optimum parameter parameter combination 5(200 kHz) - 5(50 mm/s) - 5 (5), the through-hole can also be obtained in the case of only increasing RNC compared with the parameter parameter combination 5-5-4. In the range analysis of the orthogonal experiment, the larger the value of the range R, the greater the influence of the process parameters on the evaluation index. Comparing the values of R_1_, R_2_ and R_3_, it is known that RNC, CS and PRF have an increased influence on the roundness of micro-hole in this orthogonal experiment.

The interactive diagram obtained through the interactive analysis can show how the relationship between a parameter parameter and an evaluation indicator depends on the value of the second parameter parameter. Figure 4b shows the interactions between each parameter parameter for roundness. The interaction analysis between the process parameters corresponding to the optimal roundness (200 kHz-50 mm/s-4) shows that (1) when the PRF level is 200 kHz, the interaction between CS and PRF is smaller than other PRF levels, and the interaction is smallest when the CS level is 50 mm/s. (2) When the PRF level is 200 kHz, the interaction between RNC and PRF is small compared to other PRF levels, but the interaction is greatest when the RNC level is 5. (3) When the CS level is 50 mm/s, the interaction between RNC and CS is smaller than the other CS level, and when the RNC level is 5, the interaction is larger than other RNC levels.

When the evaluation index is the diameter of the micro-hole, the results of the range analysis of each parameter parameter are shown in Table 6 and Figure 5a: (1) The parameter parameter PRF has a range R_1_ of 5.932 μm, and the optimum parameter parameter level is 4 (190 kHz); (2) the parameter parameter CS has a range R_2_ of 4.608 μm, and the optimum parameter parameter level is 4 (40 mm/s); (3) the parameter parameter RNC has a range R_3_ of 1.680 μm, and the optimum parameter parameter level is 3 (3). As shown in Table 4, the through-hole can be obtained when the parameter parameter combination is 5 (200 kHz) - 4 (40 mm/s) - 3 (3). Therefore, for the optimum parameter parameter combination 4 (190 kHz) - 4 (40 mm/s) - 3 (3), the through-hole can also be obtained in the case of only reducing the output power compared with the parameter parameter combination 5-4-3. Comparing the values of R_1_, R_2_ and R_3_, it is known that RNC, CS and PRF have an increased influence on the diameter of micro-hole in this orthogonal experiment. For the combination of process parameters (190 kHz - 40 mm/s - 3) corresponding to the optimal diameter of the micro-hole, it can be known from the interaction analysis in Figure 5b: (1) when the PRF level is 190 kHz, the interaction is smallest when the CS level is 40 mm/s. (2) When the PRF level is 190 kHz, the interaction is relatively small when RNC level is 3. (3) When the CS level is 40 mm/s, the interaction is middle when the RNC level is 3.

Similarly, when the evaluation index is the size of the HAZ, the results of the range analysis of each parameter parameter are shown in Table 7 and Figure 6a: (1) the parameter parameter PRF has a range R_1_ of 1.157 μm, and the optimum parameter parameter level is 4 (190 kHz); the parameter parameter CS has a range R_2_ of 0.637 μm, and the optimum process parameter level is 4 (40 mm/s); the parameter parameter RNC has a range R_3_ of 0.047 μm, and the optimum parameter parameter level is 3 (3). As shown in Table 4, the through-hole can be obtained when the parameter parameter combination is 5 (200 kHz) - 4 (40 mm/s) - 3 (3). Therefore, for the optimum parameter parameter combination 4 (190 kHz) - 4 (40 mm/s) - 3 (3), the through-hole can also be obtained in the case of only reducing the output power compared with the parameter parameter combination 5-4-3. Comparing the values of R_1_, R_2_ and R_3_, it is known that RNC, CS and PRF have an increased influence on the diameter of micro-hole in this orthogonal experiment. For the combination of process parameters (190 kHz - 40 mm/s - 3) corresponding to the optimal size of HAZ, it can be known from the interaction analysis in Figure 6b: (1) when the PRF level is 190 kHz, the interaction is smallest when the CS level is 40 mm/s. (2) When the PRF level is 190 kHz, the interaction is relatively small when RNC level is 3. (3) When the CS level is 40 mm/s, the interaction is relatively large when the RNC level is 3.

According to the range analysis results of the orthogonal experiment, the effect of each parameter parameter on the diameter of micro-hole and HAZ was approximate, and the minimum diameter and HAZ were obtained at the same parameter level combination of 4(190 kHz) - 4(40 mm/s) - 3(3). As shown in Table 5, the roundness corresponding to each parameter parameter at this parameter level was 0.810 μm (PRF), 0.791 μm (CS), 0.976 μm (RNC), respectively. In the combination of parameter level to obtain the best roundness, the roundness corresponding to each parameter parameter was 0.692 μm (PRF), 0.667 μm (CS), 0.869 μm (RNC). Although the parameter level combination (5-5-5) that achieves the best roundness was inconsistent with this parameter level combination (4-4-3), the roundness difference of each parameter parameter was more than 0.13 μm. Therefore, parameter level combination of 4(190 kHz) - 4(40 mm/s) - 3(3) was selected as the optimal parameter level combination of laser micro-cutting in this study. It should be emphasized that in this parameter parameter combination, the interaction of PRF-CS and PRF-RNC is small, and the interaction of CS-RNC is large.

### 3.3. Mechanism of Laser Micro-Cutting

In order to determine how the process parameters affect the quality of the micro-holes, it is important to understand the interaction mechanism between the nanosecond UV laser and the ultra-thin PI film. Figure 7 shows the typical morphology of the micro-hole before and after cleaning. As shown in Figure 7a, after the laser micro-cutting, a circular area having a diameter of several tens of micrometers is formed through the PI film, the material in the micro-hole is not completely removed and a layered halo appears around this circular area. As shown in Figure 7b, the layered halo around the micro-hole disappeared after cleaning and the material in the micro-hole was removed, indicating that by-products of laser cutting could be removed.

Further, the product of the laser micro-cutting was analyzed by Raman spectra and energy dispersion spectrum (EDS). Before cleaning, as shown in Figure 7a, the A-D points were selected from the inside of the micro-hole to the periphery of the layered halo for Raman analysis. There are no other peaks except the D band and the G band of point A at 1356 and 1594 cm^−1^ (Figure 8a), which indicates that the main component in the micro-hole after laser micro-cutting is carbon [35,36]. For the Raman spectra from point B to point D, there are no obvious peaks nor carbon peaks, which indicates that there are no new chemical bonds generated in the layered halo, and the carbon is not sputtered out from the micro-hole. According to the EDS analysis of these points, as shown in Table 8, the carbon content (97.99%) is much higher than the other positions, which further confirms the above-mentioned laser carbonization process.

After cleaning, as shown in Figure 7b, the point A_1_ on the inner wall of the micro-hole, the point B_1_ on the HAZ, and the point C_1_ outside the HAZ were selected for Raman and EDS analysis. There are no obvious peaks or carbon peaks on the Raman curve in Figure 8b, indicating that the composition of each position is basically the same after cleaning. Points A_1_, B_1_, and C_1_ have little difference in elemental content (Table 8), further indicating that there is no significant difference in the material surrounding the micro-hole after cleaning.

As shown in Figure 9, to analyze the dynamic change course of PI during laser heating, thermogravimetric analysis was performed on this ultra-thin PI film. According to the TGA curve, the plateau period is within 460 °C, and then the reaction period begins. Moreover, the thermal degradation process of PI film, a thermosetting material, is mainly one-step degradation. Thus, the weight loss step can be attributed to the carbonization process of the sample. As shown in the DTA curve, the temperature difference between the sample and the reference during heating (ΔT) is always less than zero, indicating that the PI film continues to absorb heat during thermal decomposition. The DTA curve has a peak at 619 °C, when the heat absorption rate is the highest and the weight loss rate is also the highest. In combination with the analysis of Figure 7 and Figure 8 and Table 8, PI is carbonized at high temperature and remains on the inner wall, and the short-chain polyimide component is sputtered outside the micro-hole. In addition, because of thermal diffusion, wrinkles as shown in Figure 2 are produced around the micro-hole.

With heating-carbonization as the core, the influence of the laser micro-cutting process parameters on these evaluation indexes was discussed. The nanosecond UV laser carbonizes the PI film by high temperature on a circular cutting path to form a micro-hole, and the carbonized material remains in the micro-hole. Short-chain polyimide molecules that are partially decomposed but not yet charred will sputter on the outside of the micropores to form a halo. These products can be removed by cleaning, and a wrinkled HAZ formed by thermal diffusion can be clearly observed. Under the conditions of other process parameters unchanged: (1) the higher the PRF value, the lower the output power and the smaller the carbonization diameter of the spot; (2) the faster the CS, the less heat accumulation effect, and the smaller the carbonization diameter of the spot; (3) the more RNC, the more times the material is removed by the heat accumulation effect, and the larger the carbonization area. Experiment results of roundness analysis shows that lower output power and higher CS are beneficial to reduce the heat accumulation effect, which helps to improve the roundness of the micro-holes. In addition, more RNC is conducive to sufficient carbonized material in the same path, thus improving the roundness of the micro-hole too. Lower output power and faster cutting speeds help to form smaller carbonization diameter of the spot, resulting in smaller micro-hole diameters, but may also result in the inability to carbonize the material. The best process parameters should be achieved with a small carbonization diameter and fast cutting speeds, while orthogonal experiments help to quickly screen out the best combinations (4-4-3). At the same time, the smaller the diameter of the carbonized region, the smaller the area affected by thermal diffusion (HAZ), as shown in the experiment results of diameter and HAZ analysis of the micro-holes.

### 3.4. Fabrication of the Ultra-Thin PI Film Contact Spacer

As shown in Figure 10, an ultra-thin PI contact spacer with an average circle center distance of 50 μm was fabricated using the optimal parameter parameter combination (4-4-3). In the randomly selected 10 samples, the average and variance of the roundness, diameter, and HAZ are 0.6 ± 0.3, 24.3 ± 2.3, 1.8 ± 1.1 μm. The SEM morphology shows that the micro-holes are evenly distributed on the PI film, and there are no obvious defects, such as non-through holes and irregular shapes, etc.

## 4. Conclusions

The contact spacer is the core component that regulates the measurement range and sensitivity of tactile sensors. Using nanosecond UV laser commonly used in industry, the interaction mechanism between laser and ultra-thin PI film was first studied, and the influence of laser process parameters on micro-cutting quality was explored by an orthogonal experiment. Finally, the ultra-thin PI film contact spacer was successfully fabricated, which laid the foundation for the industrial production of the tactile sensor. Specific conclusions are as follows:

The high temperature generated by the spot carbonizes the PI film and partially stays in the micro-hole. The short-chain polyimide component is sputtered outside the micro-hole during the laser micro-cutting. Thermal diffusion during laser micro-cutting causes wrinkles around the micro-hole.

In the orthogonal experiment of this study, with the increase of PRF, CS and RNC values, the circularity of micro-hole was gradually optimized, and the optimum roundness was obtained at a parameter level of 5(200 kHz) - 5(50 mm/s) - 5 (5).

In the orthogonal experiment of this study, the effect of each parameter parameter on the diameter of micro-hole and HAZ was approximate. The minimum diameter (24.3 ± 2.3 μm) and HAZ (1.8 ± 1.1 μm) were obtained at the same parameter level of 4(190 kHz) - 4(40 mm/s) - 3(3).

## Figures and Tables

**Figure 1 micromachines-12-00908-f001:**
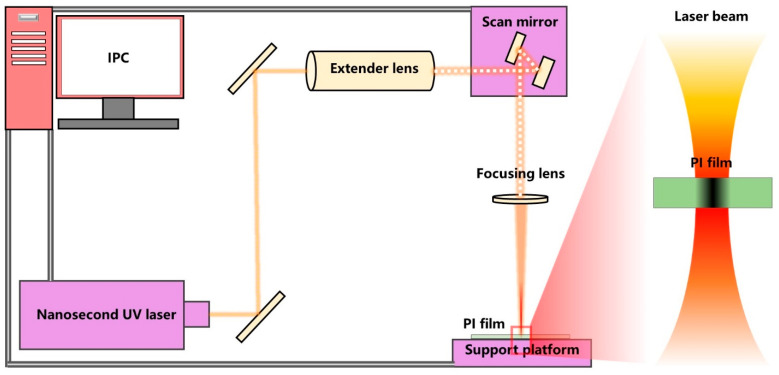
Experimental setup of PI film cutting process.

**Figure 2 micromachines-12-00908-f002:**
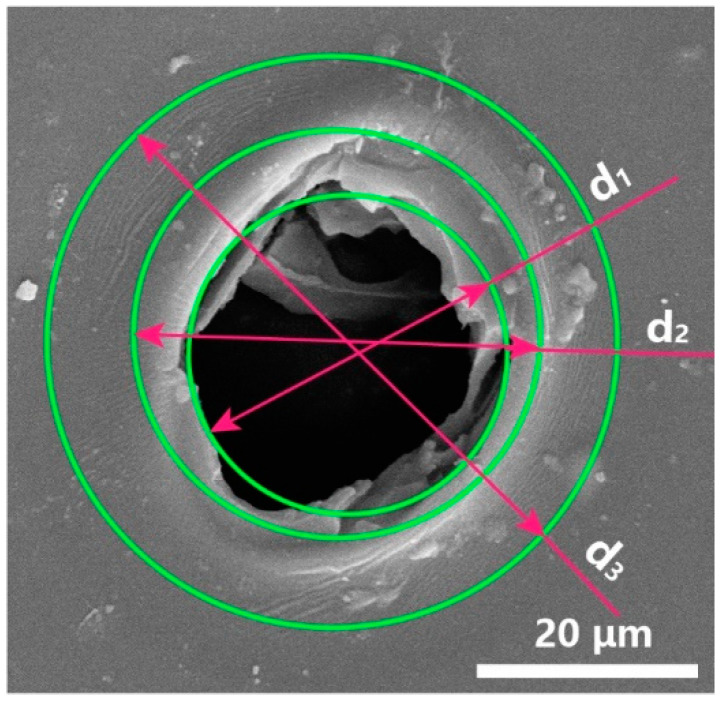
Schematic diagram of laser cutting quality evaluation.

**Figure 3 micromachines-12-00908-f003:**
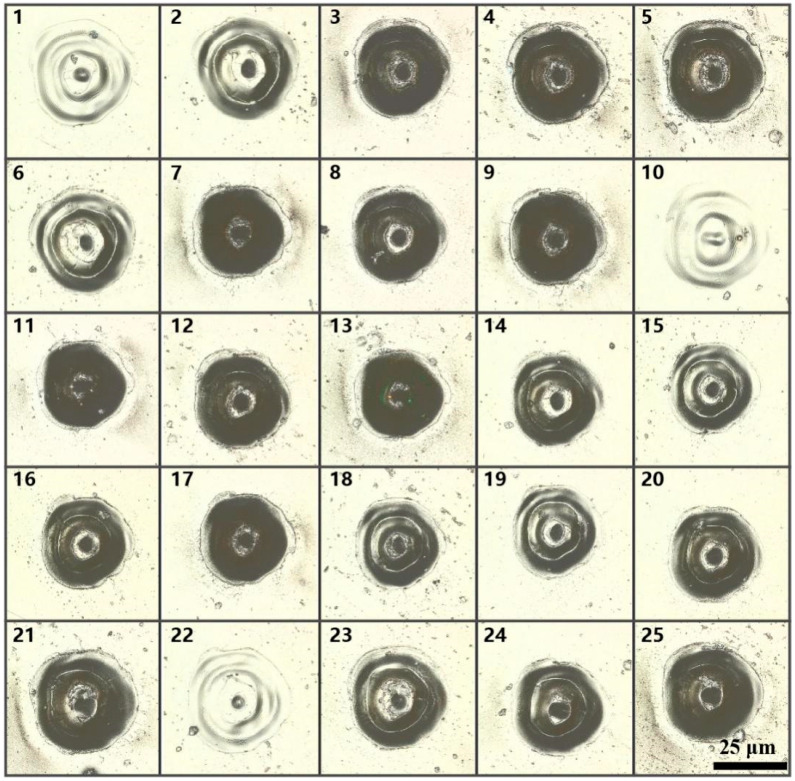
Typical confocal microscope image of each sample.

**Figure 4 micromachines-12-00908-f004:**
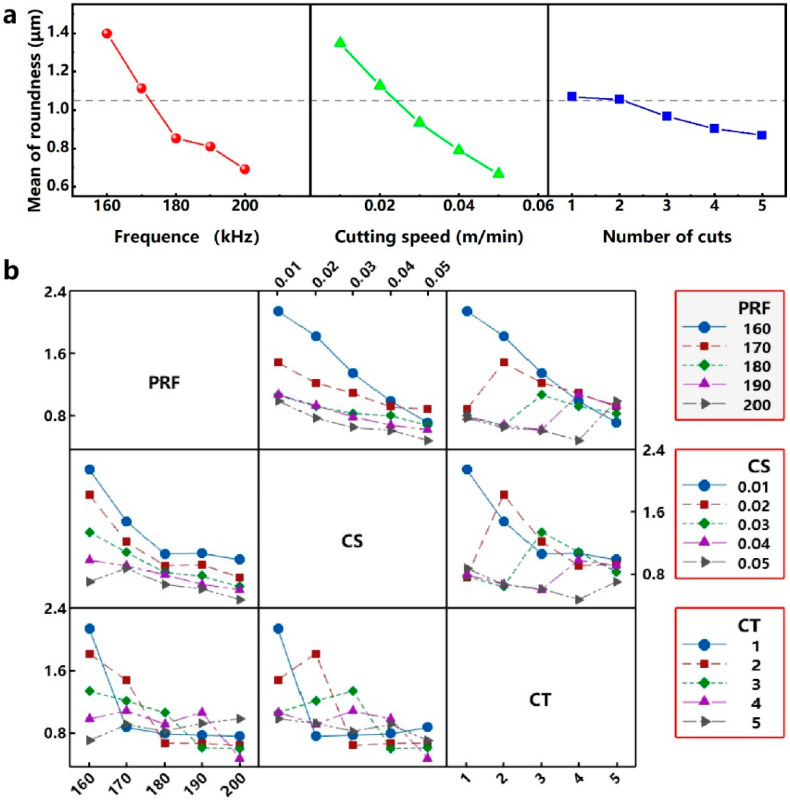
(**a**) Effect of process parameters on roundness. (**b**) Interactions between each parameter parameter for roundness.

**Figure 5 micromachines-12-00908-f005:**
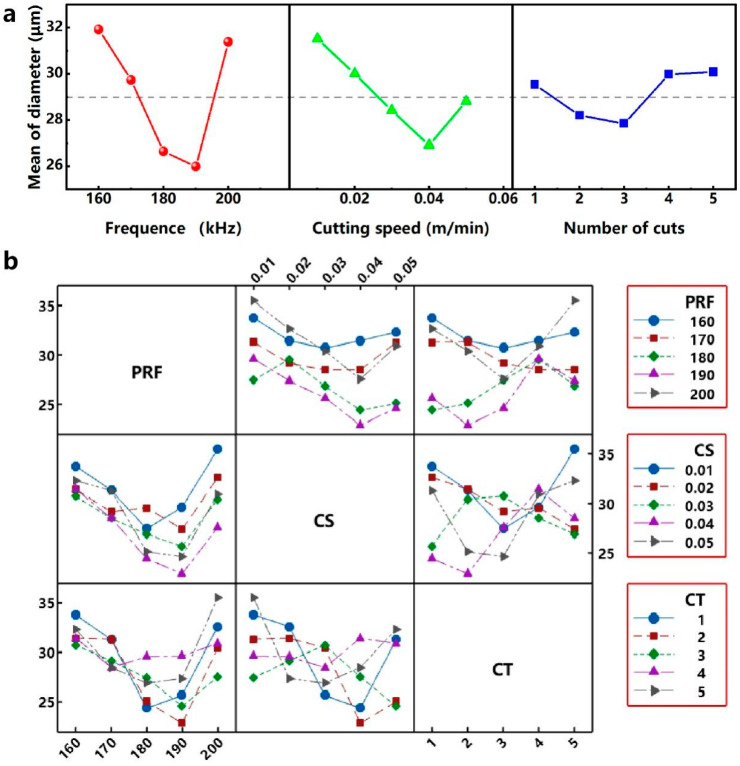
(**a**) Effect of process parameters on diameter of the micro-hole. (**b**) Interactions between each parameter parameter for diameter of the micro-hole.

**Figure 6 micromachines-12-00908-f006:**
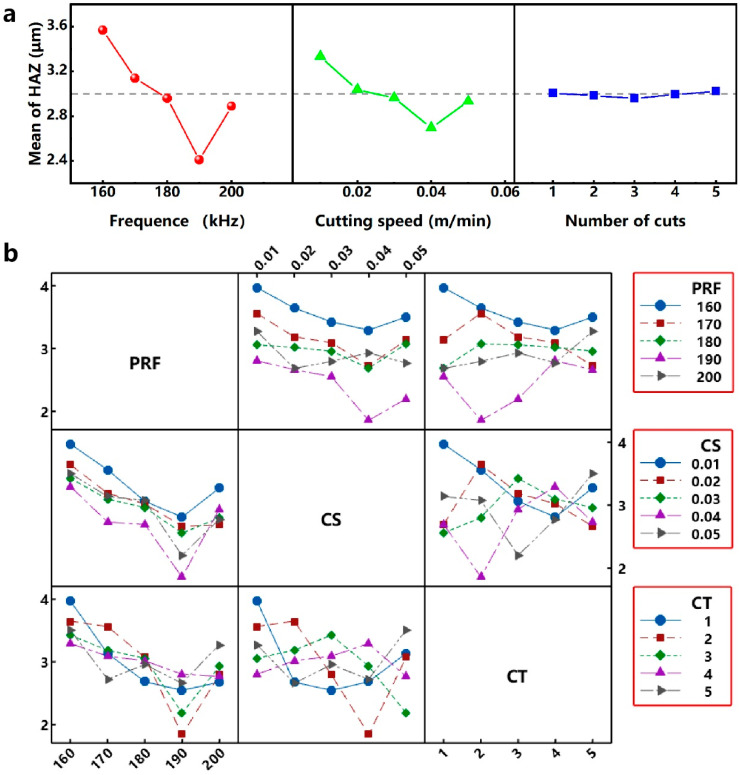
(**a**) Effect of process parameters on size of HAZ. (**b**) Interactions between each parameter parameter for size of HAZ.

**Figure 7 micromachines-12-00908-f007:**
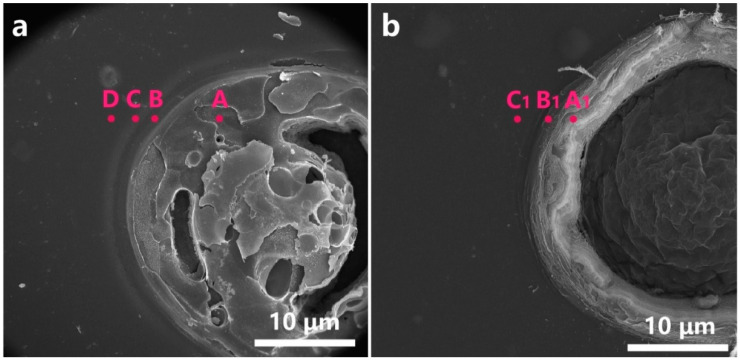
SEM images of the micro-hole (**a**) before cleaning and (**b**) after cleaning.

**Figure 8 micromachines-12-00908-f008:**
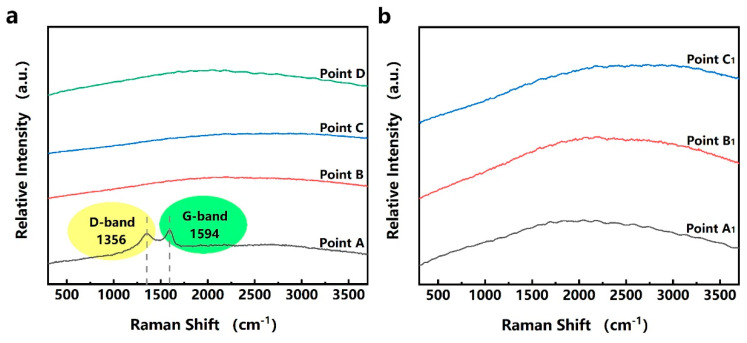
Raman measurements in different points in (**a**) Figure 7a and (**b**) Figure 7b.

**Figure 9 micromachines-12-00908-f009:**
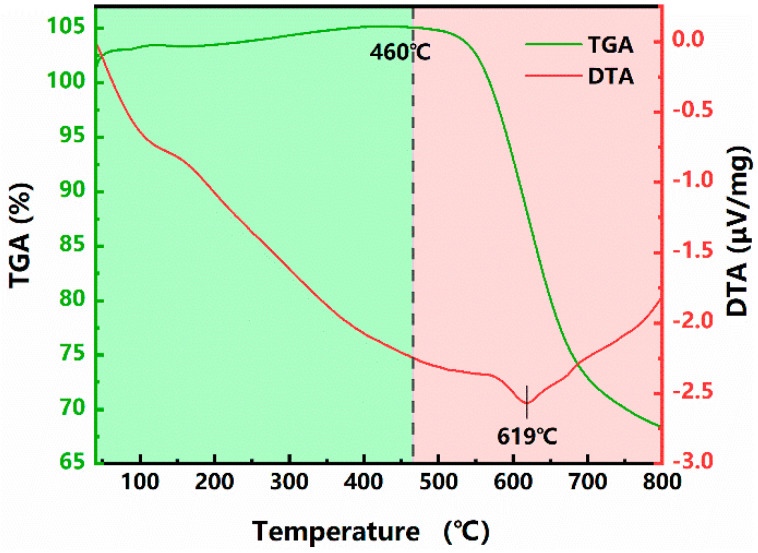
TGA and DTA curves of the ultra-thin PI film.

**Figure 10 micromachines-12-00908-f010:**
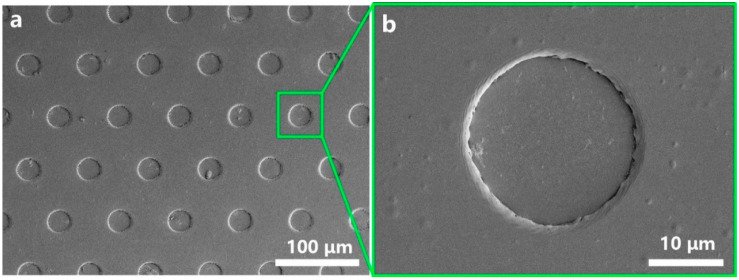
(**a**) Low- and (**b**) high-magnification SEM images of the ultra-thin PI film contact spacer.

**Table 1 micromachines-12-00908-t001:** Correspondent relationship between input PRF and output power.

Frequency (kHz)	100	120	140	160	180	200
Laser output power (W)	16.1	14.1	12.3	10.3	9.5	9.0
Actual cutting power (W)	0.159	0.137	0.121	0.102	0.94	0.89

**Table 2 micromachines-12-00908-t002:** Micro-cutting process parameters and levels.

Process Parameters	Unit	Notation	Factor Levels				
			1	2	3	4	5
Pulse repetition frequency	kHz	PRF	160	170	180	190	200
Cutting speed	mm/s	CS	10	20	30	40	50
Repetition number of cuts	——	CT	1	2	3	4	5

**Table 3 micromachines-12-00908-t003:** L_25_(5^3^) orthogonal experiments.

NO.	A	B	C	PRF	CS	RNC
1	1	1	1	160	0.01	1
2	1	2	2	160	0.02	2
3	1	3	3	160	0.03	3
4	1	4	4	160	0.04	4
5	1	5	5	160	0.05	5
6	2	1	2	170	0.01	2
7	2	2	3	170	0.02	3
8	2	3	4	170	0.03	4
9	2	4	5	170	0.04	5
10	2	5	1	170	0.05	1
11	3	1	3	180	0.01	3
12	3	2	4	180	0.02	4
13	3	3	5	180	0.03	5
14	3	4	1	180	0.04	1
15	3	5	2	180	0.05	2
16	4	1	4	190	0.01	4
17	4	2	5	190	0.02	5
18	4	3	1	190	0.03	1
19	4	4	2	190	0.04	2
20	4	5	3	190	0.05	3
21	5	1	5	200	0.01	5
22	5	2	1	200	0.02	1
23	5	3	2	200	0.03	2
24	5	4	3	200	0.04	3
25	5	5	4	200	0.05	4

**Table 4 micromachines-12-00908-t004:** Experimental results L_25_(5^3^) orthogonal experiments.

NO.	PRF (kHz)	CS (mm/s)	RNC	Roundness (μm)	Diameter (μm)	Width of HAZ (μm)	TH
1	160	10	1	2.143	33.794	3.972	No
2	160	20	2	1.822	31.410	3.646	No
3	160	30	3	1.342	30.707	3.426	Yes
4	160	40	4	0.980	31.419	3.295	Yes
5	160	50	5	0.705	32.304	3.503	Yes
6	170	10	2	1.480	31.314	3.562	No
7	170	20	3	1.217	29.144	3.187	Yes
8	170	30	4	1.085	28.474	3.093	Yes
9	170	40	5	0.907	28.481	2.725	Yes
10	170	50	1	0.875	31.279	3.136	No
11	180	10	3	1.062	27.393	3.059	Yes
12	180	20	4	0.915	29.560	3.018	Yes
13	180	30	5	0.825	26.846	2.959	Yes
14	180	40	1	0.793	24.364	2.688	No
15	180	50	2	0.668	25.051	3.074	No
16	190	10	4	1.068	29.613	2.806	Yes
17	190	20	5	0.923	27.342	2.662	Yes
18	190	30	1	0.778	25.649	2.551	No
19	190	40	2	0.670	22.806	1.848	No
20	190	50	3	0.612	24.565	2.191	No
21	200	10	5	0.985	35.492	3.272	Yes
22	200	20	1	0.760	32.618	2.681	No
23	200	30	2	0.640	30.433	2.797	Yes
24	200	40	3	0.603	27.494	2.930	Yes
25	200	50	4	0.473	30.883	2.772	Yes

**Table 5 micromachines-12-00908-t005:** Range analysis table based on roundness evaluation.

	PRF (kHz)	CS (mm/s)	RNC
$\bar{K1}$	1.398	1.348	1.070
$\bar{K2}$	1.113	1.127	1.056
$\bar{K3}$	0.853	0.934	0.967
$\bar{K4}$	0.810	0.791	0.904
$\bar{K5}$	0.692	0.667	0.869
Optimal level	5	5	5
R_j_	0.706	0.681	0.201
Order of range	R_1_ > R_2_ > R_3_		

**Table 6 micromachines-12-00908-t006:** Range analysis table based on diameter of the micro-hole.

	PRF (kHz)	CS (mm/s)	RNC
$\bar{K1}$	31.927	31.521	29.541
$\bar{K2}$	29.738	30.015	28.203
$\bar{K3}$	26.643	28.422	27.861
$\bar{K4}$	25.995	26.913	29.990
$\bar{K5}$	31.384	28.817	30.093
Optimal level	4	4	3
R_j_	5.932	4.608	1.680
Order of range	R_1_ > R_2_ > R_3_		

**Table 7 micromachines-12-00908-t007:** Range analysis table based on size of the HAZ.

	PRF (kHz)	CS (mm/s)	RNC
$\bar{K1}$	3.568	3.334	3.005
$\bar{K2}$	3.140	3.039	2.985
$\bar{K3}$	2.960	2.965	2.959
$\bar{K4}$	2.412	2.697	2.996
$\bar{K5}$	2.890	2.935	3.024
Optimal level	4	4	3
R_j_	1.157	0.637	0.047
Order of range	R_1_ > R_2_ > R_3_		

**Table 8 micromachines-12-00908-t008:** The atomic percentage in the designated area in Figure 7 before and after cleaning.

Atomic Percentage	C	O	N
Point A	97.99	0.86	1.15
Point B	73.37	19.51	7.12
Point C	81.88	12.93	5.19
Point D	71.64	20.94	7.42
Point A_1_	73.44	19.38	7.18
Point B_1_	70.87	20.69	8.44
Point C_1_	72.58	19.56	7.86

## Data Availability

Not applicable.

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
