# Peer review of "PI Film Laser Micro-Cutting for Quantitative Manufacturing of Contact Spacer in Flexible Tactile Sensor"

_micromachines, 2021, doi:10.3390/mi12080908_

Round 1

Reviewer 1 Report

The manuscript requires intensive polishing for readership. The first part (3.1) of the result section is important as the rest of the result section is following the experimental structure of 3.1, however the authors did not succeed to make clear interpretation of their experimental result, thus the rest parts are not easily understood. 

  • Reference is needed
    • Nanosecond lasers are widely used in industrial production due to their low cost and high stability. Ultraviolet (UV, 355 nm) laser has lower thermal ablation effect on the substrate than other wavelengths of light, so it is more suitable for manufacturing of organic materials to obtain smaller heat affected zone (HAZ).
  • The experiment description part must be completely rewritten. It is hardly understandable. The paragraph in 2.1 has a few duplicated sentences explaining same setting elements.
  • ‘Cutting time’ (CT) should be defined clearer. Does it mean the repetition number of the cutting or literally ‘time’ making a round?
  • In 3.1 for quality evaluation, how did the authors determine the three different diameters d1, d2, and d3? What is the definition of them? It does not seem to be clear to the reviewer. What was the morphological definition of HAZ? What kind of evaluation method was used to cross-check?
  • Notations should be defined and used as they are defined. In table 4, what is R, D, S? Are they Rou, Dia, Size in page 5?
  • The authors used PI film attached on a stainless steel. I wonder how they determined through hole from Figure 3 and Table 4? For example, #20 and #23 look alike, however one is labeled as blind hole, and the other as through hole.

Format, grammar, English should be rigorously corrected.

  • English must be seriously improved including sentences below for examples.
      • As a core component of flexible tactile sensors, contact spacer can regulation performance by adjusting contact area changes.
      • In this paper, an ultra-thin micro-hole polyimide (PI) film contact spacer was successfully fabricated using nanosecond ultraviolet (UV) lase.
      • Under this optimal laser craft parameters, the minimum diameter and HAZ were 24.3 μm ± 2.3 μm and 1.8 μm ± 1.1 μm, respectively.
      • However, the evaluation indexes of micro-hole quality, the influence of craft parameters on evaluation indexes, and the interaction mechanism between UV laser and PI film remain to be studied.
      • Secondly, the influences of craft parameters on the evaluation indexes were studied by an orthogonal experiment of 3 factors and 5 levels, and the optimal PI film laser micro-cutting craft parameters were obtained.

Reviewer 2 Report

  1. You presented a lot of literature about contact spacers and their use and fabrication, but literature about laser ablation of your material is missing. A lot of previous work on this material exists and should be referenced. Also, without having these references and without a small summary of the state of the art, I cannot estimate the novelty of your approach. A section about the state of the art of laser ablation of PI materials is required, followed by a small statement of how your work contributes to this topic.
  2. I don't understand your orthogonal 5^3 parameter screening. Having 3 parameters and 5 values each, gives 125 for full factorial sampling. I don't understand, how you found the best parameters with your kind of sampling. Please explain better.
  3. The explanations in the last paragraph of 3.3 are a little but unclear or missleading. Don't use "worse" along with heat accumulation and larger cutting speed, because smaller HAZ is usually better.
  4. You use a lot of unusual terms, please check if these are correct or find better terms:
    1. "craft" parameters: "craft" is not used in scientific context
    2. 3 factors 5 levels: uncommon
    3. cutting "times": Number of passes or repetitions
    4. "wrinkles": Is this the correct technical term?
    5. "throgh-hole": full penetration
  5. Please check your grammar. You often use singular without using an article. Use either plural or add articles (in most cases). Not required for indifinite cases like "after laser processing".
  6. See additional comments on small corrections in the uploaded PDF.

Reviewer 3 Report

The article describes the parametric study of micromachining of polyimide film using nanosecond UV laser pulses. An impressive number of samples were performed and analyzed. As the result, the optimal set of experimental parameters was found. However, the description of the experiment is incomplete. There is a lack of the most important parameters like laser beam geometry. If we do not know the beam diameter, beam divergence, focus diameter it is difficult to estimate what was the energy fluence with a given set of parameters. And without this information, the whole further analysis is pointless. 

Other remarks:

  1. In the Experimental description, there is information that the role of "extender lens" is to reduce laser processing power. Is it true? So, how it works? Or, maybe it is a laser beam expander that is used to increase laser beam diameter and reduce divergence to obtain a smaller laser focus?
  2.  In Table 1 there is "Actual cutting power" given which seems like "Laser output power " divided by 100, roughly. Why is that? And how was measured or calculated? 
  3. There is a number of editorial errors like RPF instead of PRF, lack of space before brackets, or even part of text like on page 2, line 42. 

Author Response

Thanks for your suggestion and your consideration is very comprehensive. Every comment and suggestion you provide is of great significance to the further modification of our manuscript and subsequent research work. Please see attached for detailed responses

Round 2

Reviewer 2 Report

Thank you for considering my comments. All open questions are answered by the authors and I have now further comments before publication.

Author Response

Thanks for your suggestion. Every comment and suggestion you provide is of great significance to the further modification of our manuscript and subsequent research work. 

Reviewer 3 Report

Sill, there is a lack of beam divergence and focus diameter, and energy fluence. 

Author Response

Reviewer #3:

Still, there is a lack of beam divergence and focus diameter, and energy fluence. 

Author reply:

Thanks for your suggestion and your consideration is very comprehensive. According to your suggestion, we have added beam geometry, beam diameter and other information in the manuscript.

The detailed modifications are listed as below:

(Page 3, Line 13) Insert “In this laser processing system, the laser wavelength λ is 355 nm, the spatial mode M2≤1.3, the focal length f is 167 mm, the laser spot diameter D at the entrance port is 8.5 mm, and the calculated spot diameter C for this Gaussian beam is 11.5 μm (C=4M2λf/πD).”
